# Insight into the Interaction between Water and Ion-Exchanged Aluminosilicate Glass by Nanoindentation

**DOI:** 10.3390/ma14112959

**Published:** 2021-05-30

**Authors:** Xiaoyu Li, Liangbao Jiang, Jiaxi Liu, Minbo Wang, Jiaming Li, Yue Yan

**Affiliations:** 1Department of Transparent Materials, Beijing Institute of Aeronautical Materials, Beijing 100095, China; jiaxi.liu@biam.ac.cn (J.L.); minbowang@126.com (M.W.); jiamingli@126.com (J.L.); 2Beijing Engineering Research Center of Advanced Structural Transparencies for the Modern Traffic System, Beijing 100095, China

**Keywords:** surface hydration, ion-exchange, nanoindentation, hardness, Young’s modulus

## Abstract

This work aims to explore the interaction between water and ion-exchanged aluminosilicate glass. The surface mechanical properties of ion-exchanged glasses after different hydration durations are investigated. The compressive stress and depth of stress layer are determined with a surface stress meter on the basis of photo-elasticity theory. The hardness and Young’s modulus are tested through nanoindentation. Infrared spectroscopy is used to determine the variation in surface structures of the glass samples. The results show that hydration has obvious effects on the hardness and Young’s modulus of the raw and ion-exchanged glasses. The hardness and Young’s modulus decrease to different extents after different hydration times, and the Young’s modulus shows some recovery with the prolonging of hydration time. The ion-exchanged glasses are more resistant to hydration. The tin side is more resistant to hydration than the air side. The results are expected to serve as reference for better understanding the hydration process of ion-exchanged glass.

## 1. Introduction

Ion-exchanged glasses are used as the main material for fabricating the touch panels of popular display devices, forward-facing aircraft, and other windscreens, and their surface mechanical properties are important requirements [1,2]. The surfaces of silicate glasses hydrate when in water-containing environments [3,4,5]. This reaction starts from the production process, and continues during storage and use [6]. Hydration may cause the deterioration of the glasses and enhance the growth of surface cracks, thereby reducing the strength of the glasses. Therefore, investigating the surface mechanical properties of glasses and exploring the effects of hydration on these properties are of fundamental importance [7].

The well-known water corrosion mechanism includes hydration, hydrolysis and leaching [8,9]. Hydration, hydrolysis, and leaching usually act simultaneously during glass corrosion in water. Hydration is a penetration process whereby intact water molecules spread into the glass substrate, whereas hydrolysis and leaching are chemical reactions involving ion exchange between ions in the water and in the glass structure [10]. The accepted hydrolysis and leaching reaction equations can be described as:(1)$$\equiv \mathrm{Si}-O-\mathrm{Si}\equiv +H_{2}O\to 2\equiv Si-OH$$
(2)$$\equiv Si-O-Na+H_{2}O\to \equiv Si-OH+Na^{+}+OH^{-}$$

This process can be described by the classical stress corrosion theory developed by Charles and Hillig [11,12]. This theory states that molecules possess proton donor sites and lone-pair orbitals such that the dissociation of the Si–O–Si bond is enhanced by water molecules through coupling across the Si–O bond to form an activated complex. Hydrolysis can attack the Si–O–Si bonds directly, and this process is the principal reason for strength decrease in glasses [5]. Leaching weakens the structure stability of glass, provides voids assisting water molecular penetration, and helps in forming the hydration layer through the exchange of modifier ions. Different types of hydrated surface layers can be formed depending on the durability of the glasses. The structure of the hydrated layer is different for various types of glasses [3].

Several attempts have been made to study the influence of hydration on the mechanical properties of glasses [13,14,15,16]. The wear volume of fused quartz glass gradually increases with the increase in environment humidity; however, the wear tracks on soda lime glass are smoother than those on fused quartz [13]. The surface wear of soda lime glass decreases with the increase in humidity, whereas that of sodium aluminosilicate glass remarkably increases [14]. Most of the studies focus on the variation in the mechanical properties of the glasses, whereas studies on the mechanical properties of hydrated surfaces (especially at extremely low depths) are limited [3,7,14,17]. Recent results obtained by nanoindentation show that the near-surface mechanical properties are altered by hydration, and the glass composition, determining the resistance of the glass, is attacked with an aqueous solution. Hydration shows minimal effects on high-durability glasses, even at long immersion times, whereas hydration reduces the near-surface mechanical properties of low-durability glasses remarkably [3]. Surface hydration also influences the residual compressive stress (CS) in ion-exchanged glasses [5,18]. Alkali modifier ions on the surface of ion-exchanged glasses are leached when the ion-exchanged glasses are in contact with water, regardless of the surface compressive layer [3]. However, the influence of hydration on the surface nano-mechanical properties of ion-exchanged aluminosilicate glass is still unknown.

During the float process, the molten glass is deposited on a tin bath to obtain a flat parallel surfaces [19,20,21]. The tin ions can penetrate into the bottom surface of the float glass and result in two chemically different sides, which are often referred to as the air and tin sides. The difference in composition and structure between the two sides can lead to different properties [4,22,23,24,25]. The diverse performance between the air and tin sides of ion-exchanged glasses in the water hydration process is still little understood.

This study mainly aims to investigate the influence of hydration on the surface mechanical properties of ion-exchanged aluminosilicate float glass. The surface hardness and Young’s modulus of hydrated and non–hydrated glasses are obtained through nanoindentation. The variation in surface structures is explored through infrared (IR) spectroscopy.

## 2. Materials and Methods

The glass used was a float glass provided by AVIC SANXIN. The glass composition was the same as that in our previous published research [23,25,26], as shown in Table 1. The square plate glass samples measured an average of 50 ± 0.5 mm on an edge and an average thickness of 4.0 ± 0.02 mm. Ion exchange involved suspending the aluminosilicate glasses in a molten potassium nitrate bath held at 420 °C for different times (1 and 12 h). The glass samples were then removed, cooled, and rinsed with deionized water after ion exchange.

The samples were hydrated by immersion in distilled water at 60 °C for various time periods (5, 10, 15, and 20 days). The samples were kept vertical during immersion. At the end of immersion, the samples were removed from the water and gently dried with a warm air blower to prevent the removal of the hydrated layer. All prepared samples were sealed in a vacuum to prevent contamination on the surface until all the tests were completed.

The CS and depth of stress layer (DOL) of the glass samples were measured with a surface stress meter (FSM-6000LE, ORIHARA, Japan) on the basis of photoelasticity theory [27]. The glass sample was placed on the measuring area, and care was taken to guarantee the glass surface fit with the triple prism so as to ensure the birefringence fringes were detectable and clear. Parameters such as the thickness, refractive index, and photoelastic constant of the glass were inputted in the analysis [16]. The refractive index (590 nm) for the compression layer measurement was 1.51, and the photoelastic constant was 27.0. The CS and DOL values were then obtained. The systematic error for the CS was ± 20 MPa, and for the DOL was ± 2 μm. Each glass sample was measured at five random positions, and the average CS and DOL values were obtained.

Nanoindentation was conducted with an XP nanoindenter equipped with a Berkovich diamond indenter (Hysitron, TI 950). Prior to the nanoindentation tests, calibration was performed using a standard fused silica sample. The indentation load changed from 100 μN to 9000 μN in a proportional sequence, with a total of 36 indents. The Poisson’s ratio of the sample used in the calculation was 0.25. The method used in the experiments and calculations is based on ISO 14577 [28]. The hardness and modulus data were calculated using the well-known method developed by Oliver and Pharr (referred to as the OP method). During nanoindentation tests, the temperature was maintained at 26 ± 1 °C with a relative humidity of 60–80%. The nanoindentation tests were conducted immediately after the sample was dried.

IR spectra were recorded using a Spectrum GX FTIR system (PerkinElmer) with a specular reflectance accessory in the range of 4000–400 cm^−1^ at room temperature to evaluate the surface structural changes caused by hydration. The incidence angle was 45°. Each spectrum was considered an average of 16 scans collected over the frequency range of 1300–400 cm^−1^ with a resolution of 4 cm^−1^. Three samples for each hydration time were analyzed to confirm the reproducibility of the results. A silver mirror was used as a standard reference for all measurements. All results were rectified with the Kramers–Kronig (K–K) relationship.

## 3. Results

The CS and DOL values on two sides of the ion-exchanged specimens before hydration are shown in Table 2. The CS decreases, whereas the DOL increases, with the prolonging of ion-exchange time. The tin side always shows higher values for the CS and lower values for the DOL than the air side.

The CS and DOL values of the ion-exchanged glasses as a function of immersion time are shown in Figure 1a,b, respectively. A slight decrease is observed in the CS values. The values of DOL are mostly invariant with immersion time.

Figure 2 shows the hardness as a function of indentation depth for the air and tin sides of the raw glass and ion-exchanged glasses. Figure 2a shows significant drop in the measured hardness on the air side of the raw glass after immersion in water for different periods. However, the hardness variation on the tin side of the raw glass in Figure 2b is limited with immersion time. Figure 2c–f show the hardness variation of the ion-exchanged glasses with different ion exchange times. The results are similar to that of the raw glass. The hardness at three different indentation depths (50, 100, and 150 nm) is remeasured 6 times to quantify the hardness variation, and the results are shown in Figure 3. From Figure 3, we see that the hardness of the ion-exchanged glasses was greater than that of the raw glass. The hardness decreases first with immersion time and then increases, and this trend is similar to the raw glass and ion-exchanged glasses. The hardness reduction of the ion-exchanged glasses was less than that of the raw glass, as shown in Figure 3a,b.

The Young’s modulus as a function of the indentation depth for the raw glass and the ion-exchanged glasses is shown in Figure 4. The Young’s modulus of the raw and ion-exchanged glasses showed a remarkable variation with immersion time. The Young’s modulus of the raw glass decreased to a minimum after immersion in water for 5 days and then recovered to different extents with the prolonging of the immersion time. The Young’s modulus of the ion-exchanged glass for 1 h decreased to a minimum after immersion in water for 10 days, and then showed some recovery with the prolonging of immersion time. For the ion-exchanged glass for 12 h, the Young’s modulus decreased to a minimum after immersion in water for 20 days. The reduction in Young’s modulus on the tin side of the raw and ion-exchanged glasses was less than that on the air side of the same glass, which is similar to the hardness results.

Water can exist in glass as a hydroxyl group (Si–OH) or as interstitial molecular water. A stretching vibration bond of O–H in Si–O–H species can be found at approximately 3662–3697 cm^−1^, and O–H stretching vibrations of liquid-like molecular water (H_2_O) can be found at approximately 3400 cm^−1^, shown as a broad peak [29,30,31]. Therefore, the IR spectra of 3300–4000 cm^−1^ for the raw glass and ion-exchanged glasses are obtained and shown in Figure 5, to investigate the structure variation in glass surfaces caused by hydration.

A sharp peak at 3697 cm^−1^ occurred in the IR spectra, whereas a 3400 cm^−1^ absorption band was not observed, as shown in Figure 5. The peaks are enlarged and shown as insets in the figures to provide additional details. The IR spectra on one side of the glass samples are influenced by the other side due to the transparency of the glass, and total prevention of influence is relatively difficult. However, the intensity of the IR beam from the exposed surface is stronger than that from the other side of the sample. Consequently, the IR spectra are mainly composed of light from the exposed surface [32]. All the results are rectified with the K–K relationship to avoid interference as much as possible.

From Figure 5, we can see that the absorbance of different glass samples at 3697 cm^−1^ varied with immersion time. For the raw glass, the percentage of absorbance at 3697 cm^−1^ increased to a maximum after immersion in water for 5 days, and then decreased with the prolonging of immersion time. The percentage of absorbance at 3697 cm^−1^ increased to a maximum after the immersion in water for 10 days of the ion-exchanged glass for 1 h, and increased to a maximum after the immersion in water for 20 days of the ion-exchanged glass for 12 h. The tin side showed a similar tendency as the air side in the raw and ion-exchanged glasses.

Figure 6 shows the comparison of IR spectra at 3697 cm^−1^ between the raw glass and ion-exchanged glasses with different immersion times. The percentage of absorbance at 3697 cm^−1^ for the ion-exchanged glasses was greater than that of the raw glass for the as-prepared and hydrated samples. The air side consistently showed a higher value of absorbance at 3697 cm^−1^ than the tin side in the raw and ion-exchanged glasses.

## 4. Discussion

Surface CS relaxation occurs at a fast rate in the presence of water, and this is due to the lowered viscosity of the glass surface that is affected by water [18]. However, the CS results in Figure 1 show a slight decrease with different immersion times. This condition may be because the hydronium ions in the Na^+^-leached sites can produce new CS that superposes with the original CS produced by K^+^–Na^+^ ion exchange [13], thereby leading to an extremely small change in CS.

During the corrosion of silicate glass, the in situ repolymerization of the silica network followed by the release of soluble elements can form an amorphous film on the surface of glasses [33,34]. This amorphous film is commonly referred to as a “gel” layer [35], and it is rich in the alkali-depleted silanol (Si–OH) group that can cause lower mechanical properties [3,36]. The hardness (Figure 2) and Young’s modulus (Figure 4) obtained for the as-prepared and hydrated samples do not overlap, indicating the presence of a “gel” surface layer on the latter samples.

Water can exist in glass as a hydroxyl group (Si–OH) or as interstitial molecular water. The majority of water exists as Si–OH when the total water concentration is less than 1000–3000 ppm in the bulk, whereas it exists in interstitial molecular form at higher concentrations [37]. IR spectroscopy was used to characterize the different hydrous species in glass samples. The IR absorption intensities of the O–H species were determined after baseline subtraction. As shown in Figure 5, a sharp peak standing for the stretching vibration bond of O–H in Si–O–H species can be found at 3697 cm^−1^, and the peak standing for the O–H stretching vibrations of liquid-like molecular water (H_2_O) is absent. The absence of a broad peak at approximately 3400 cm^−1^ indicates that most of the hydrogen in the glass surface existed as Si–O–H species rather than molecular water. The percentage of absorbance at 3697 cm^−1^ increased to different extents after different immersion times in the raw and ion-exchanged glasses, confirming the formation of the hydrated “gel” layer.

The decrease in hardness on the tin side was less than that on the air side in the raw and ion-exchanged glasses from Figure 3, indicating a stronger resistance to hydration of the tin side. This result is in accordance with previous work [4,38]. The presence of tin ions at the tin side and surface hydration intrinsically change the near-surface structure, as does surface hydration. Ziemath et al. [39] suggested that the high field strength of Sn^4+^ leads to a compact structure, thereby reducing the mobility of alkali ions and delaying the corrosion process. The tin ions consume the sodium ions on the tin side at a depth of approximately 100 nm [40]. The already sodium-depleted layer may delay the interdiffusion of sodium and hydronium ions and lead to slower corrosion [4]. The IR spectroscopy results (Figure 6) show that the air side of the glass samples held a larger absorbance percentage of OH peak than the tin side before and after corrosion, indicating the lower mechanical properties on the air side.

As shown in Figure 3, the reduction in hardness of ion-exchanged glasses is less than that of the raw glass, indicating that ion-exchanged glasses are more resistant to hydration. The corrosion mechanism, namely, the stress corrosion mechanism, may be different when certain stresses are applied on glasses [4]. The glass is much more chemically active under tensile stress and reacts easily with water molecules, and this condition is called stress-enhanced hydrolysis [41]. The corrosion is inhibited for ion-exchanged glasses with a great CS on the surface [16]. A higher surface density on the ion-exchanged glasses caused by the insertion of larger K^+^ ions can inhibit the penetration of water molecules [42]. The K^+^ ions on the surface of ion-exchanged glasses with a large radius and high polarizability show a lower mobility compared with the sodium ions that have been replaced [43]. These factors can lead to the lower corrosion rate of ion-exchanged glasses. From Figure 6, we see that the percentages of absorbance at 3697 cm^−1^ that represents the concentration of Si–OH species on the surface of ion-exchanged glasses are higher than in the raw glass. This result agrees with previous studies [44,45], and can be ascribed to the penetration of water impurities that exist in the molten KNO_3_ used for the ion-exchange process in the glass surface [18]. The Si–OH species can reduce the surface mechanical properties, which is not in accordance with the mechanical properties’ results. However, the CS on the surface of the ion-exchanged glasses can enhance the mechanical properties. CS may be the dominant effect in altering the surface mechanical properties. Therefore, the mechanical properties of ion-exchanged glasses are better than those of the raw glass.

Figure 4 shows that the Young’s modulus of the glasses decreases to a minimum and then recovers with the prolonging of immersion time. The hardness results in Figure 3 show a similar trend. This result is similar to a previous work [16], wherein the crack extending velocity in aluminosilicate glass increases after immersion in water for 24 h and then decreases with the prolonging of immersion time. This condition can be attributed to crack-healing effects (e.g., crack tip blunting) [46]. In the glass structure, water causes the glass to swell, and CS builds up and increases the mechanical properties because the glass structure is prevented from expanding freely [16,47]. The “gel” layer can be densified in the continuous hydration process, leading to the closure of pores in the “gel” layer [35]. This effect can lead to the recovery of the surface mechanical properties. The Young’s modulus of the ion-exchanged glasses decreases to a minimum after a longer immersion time than the raw glass, indicating a stronger resistance to hydration in the ion-exchanged glasses.

## 5. Conclusions

The surface CS slightly decreases with the prolonging of hydration time, whereas the DOL is mostly invariant in the same process.

The hardness and Young’s modulus of the as-prepared and hydrated samples do not overlap, indicating the presence of a “gel” surface layer on the latter samples. The IR spectroscopy results show that most of the hydrogen in the glass surface exists as Si–O–H species, rather than molecular water.

The decrease in hardness and Young’s modulus on the tin side is less than that on the air side for the raw and ion-exchanged glasses, indicating a stronger resistance to hydration in the tin side. This condition is attributed to the high field strength of Sn^4+^ that can lead to a compact structure, thereby reducing the mobility of alkali ions and delaying the corrosion process. The already sodium-depleted layer consumed by tin ions may delay the inter-diffusion of sodium and hydronium ions, and lead to slower corrosion.

The decrease in the mechanical properties of ion-exchanged glasses is less than that of raw glass, indicating that ion-exchanged glasses are more resistant to hydration. This is because the greater surface density on the ion-exchanged glasses can inhibit the penetration of water molecules. The larger radius and higher polarizability of K^+^ ions cause a lower mobility compared with the sodium ions that have been replaced, and this condition also inhibits hydration.

The Young’s modulus of the glasses decreases to a minimum, and then recovers with the prolonging of immersion time. This condition is ascribed to the swelling of glass caused by water penetration and the closure of pores in the “gel” layer in continuous hydration.

Touch panels of popular display devices and the latest-generation forward facing windscreens on commercial aircraft now use chemically strengthened glasses. The results in this study are expected to serve as reference for better understanding the hydration process of ion-exchanged glass.

## Figures and Tables

**Figure 1 materials-14-02959-f001:**
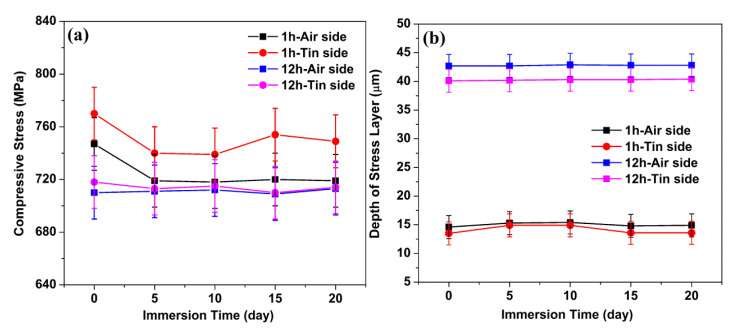
CS (**a**) and DOL (**b**) as a function of immersion time.

**Figure 2 materials-14-02959-f002:**
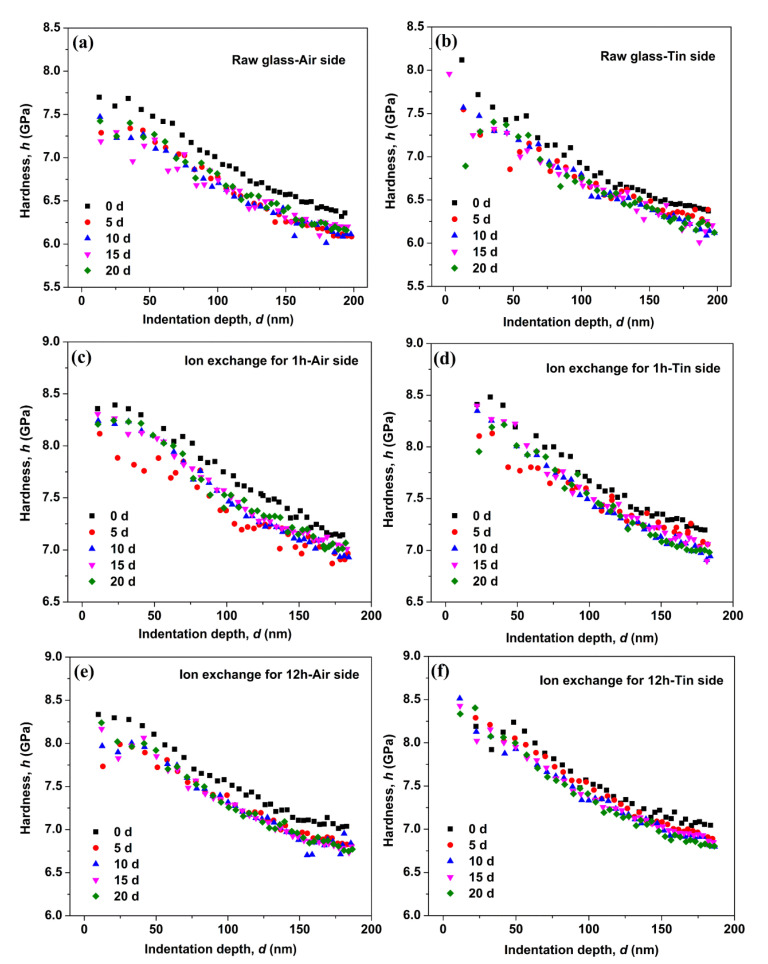
Hardness as a function of the indentation depth on the air side of the raw glass (**a**), on the tin side of the raw glass (**b**), on the air side of the ion-exchanged glasses for 1 h (**c**), on the tin side of the ion-exchanged glasses for 1 h (**d**), on the air side of the ion-exchanged glasses for 12 h (**e**) and on the tin side of the ion-exchanged glasses for 12 h (**f**).

**Figure 3 materials-14-02959-f003:**
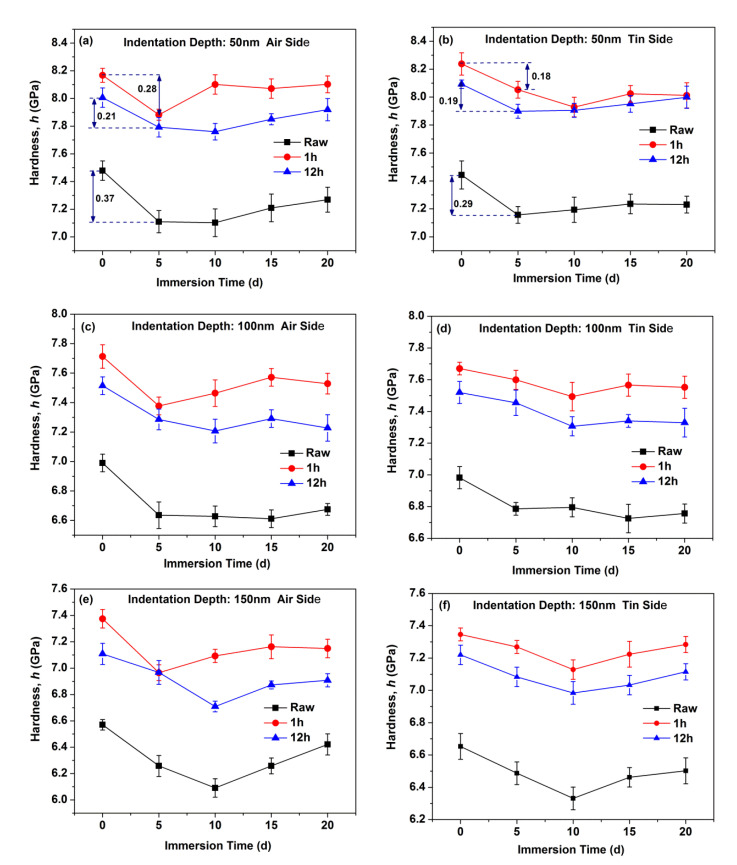
Hardness as a function of the immersion time on the air side at indentation depth of 50 nm (**a**), on the tin side at indentation depth of 50 nm (**b**), on the air side at indentation depth of 100 nm (**c**), on the tin side at indentation depth of 100 nm (**d**), on the air side at indentation depth of 150 nm (**e**), and on the tin side at indentation depth of 150 nm (**f**).

**Figure 4 materials-14-02959-f004:**
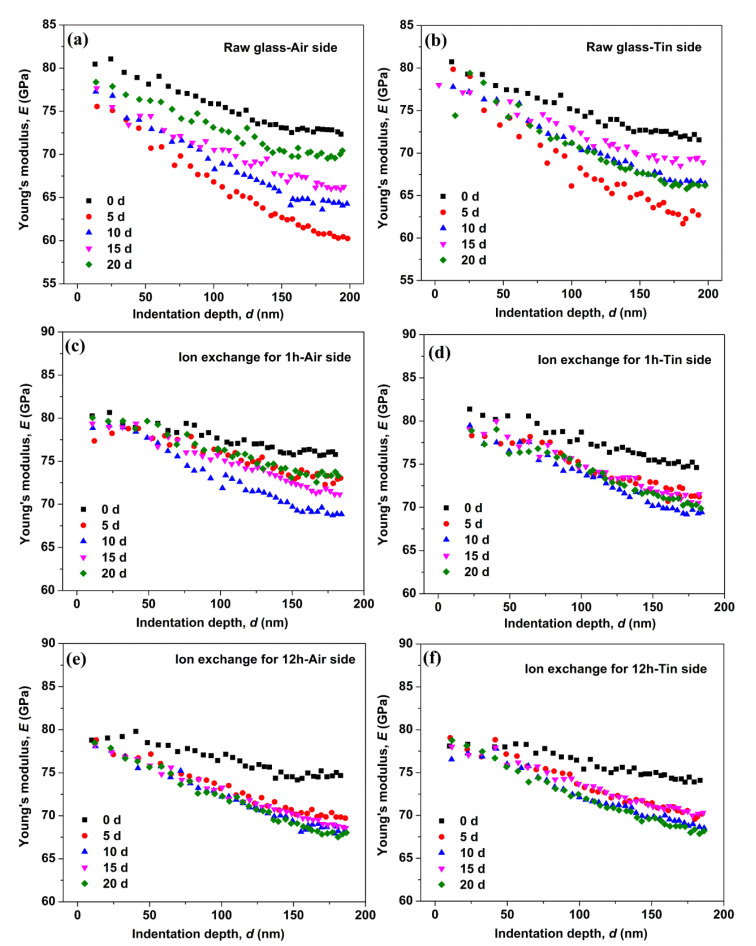
Young’s modulus as a function of the indentation depth on the air side of the raw glass (**a**), on the tin side of the raw glass (**b**), on the air side of the ion-exchanged glasses for 1 h (**c**), on the tin side of the ion-exchanged glasses for 1 h (**d**), on the air side of the ion-exchanged glasses for 12 h (**e**), and on the tin side of the ion-exchanged glasses for 12 h (**f**).

**Figure 5 materials-14-02959-f005:**
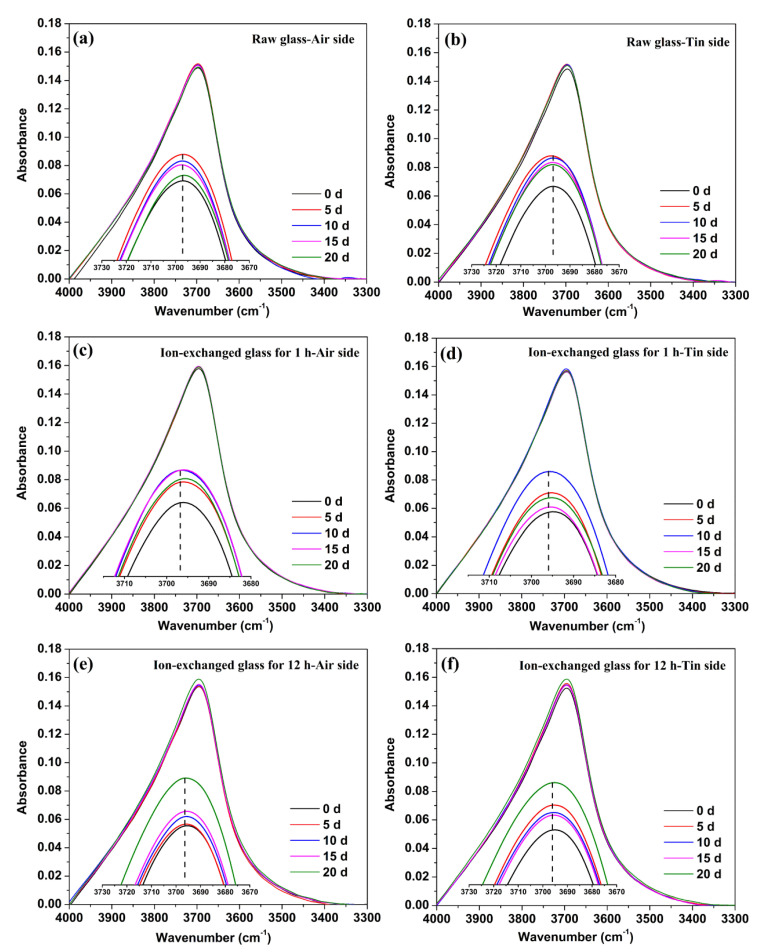
IR spectra of 3300–4000 cm^−1^ on the air side (**a**) and tin side (**b**) of the raw glass, on the air side (**c**) and tin side (**d**) of the ion-exchanged glass for 1 h, and on the air side (**e**) and tin side (**f**) of the ion-exchanged glass for 12 h. The peaks are enlarged and shown as the insets.

**Figure 6 materials-14-02959-f006:**
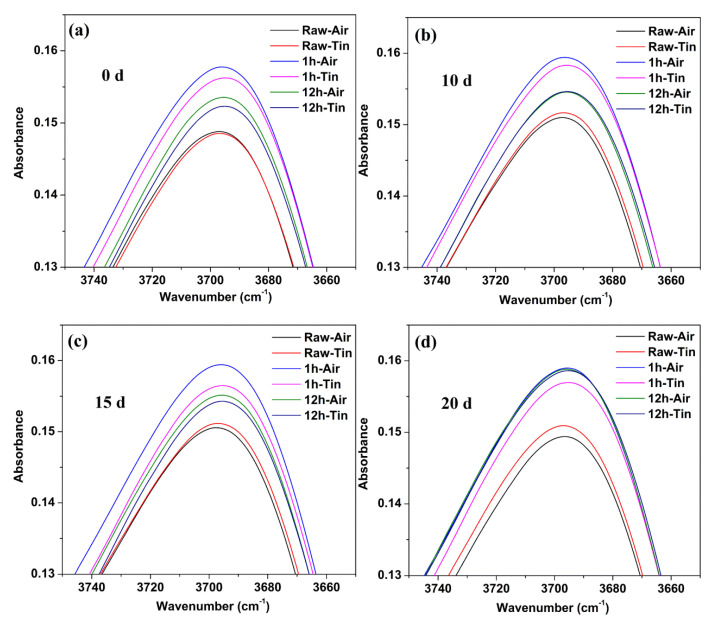
IR spectra at 3697 cm^−1^ for the raw glass and ion-exchanged glasses with different immersion times: (**a**) 0 d; (**b**) 10 d; (**c**) 15 d; (**d**) 20 d.

**Table 1 materials-14-02959-t001:** Mean chemical composition of the glass used.

Oxide	SiO_2_	Al_2_O_3_	MgO	Na_2_O	K_2_O	CaO	Fe_2_O_3_	Others
Wt. %	63.5	5.8	10.8	13.2	5.9	0.3	0.1	0.4

**Table 2 materials-14-02959-t002:** CS and DOL of the ion-exchanged glasses.

Ion Exchange Time (h)	CS (MPa)		DOL (μm)	
	Air Side	Tin Side	Air Side	Tin Side
1	747 ± 20	770 ± 20	15 ± 2	14 ± 2
12	710 ± 20	718 ± 20	43 ± 2	40 ± 2

## Data Availability

Data sharing not applicable.

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
