# Peer review of "Insight into the Interaction between Water and Ion-Exchanged Aluminosilicate Glass by Nanoindentation"

_materials, 2021, doi:10.3390/ma14112959_

Round 1

Reviewer 1 Report

The manuscript entitled “Insight into the interaction between water and ion-exchanged aluminosilicate glass by nanoindentation” submitted by Xiaoyu Li and co-workers for consideration for publication in the MDPI journal Materials presents the investigation in the area of the surface mechanical properties of ion-exchanged glasses after different hydration duration. The subject of the presented manuscript seems to be interesting. The presentation of the results needs revision. Moreover, a few questions appeared, which are listed below. I recommend major revision before possible acceptance in the MDPI journal Materials.

  1. In the whole text, ions should be presented as ions Sn4+ should be Sn4+ and many others
  2. Language presentation should be revised, English should be polished, and the part of sentences should be revised. In my opinion, the current language presentation in many cases seem to be too terse (lapidary). Part of sentences such as “It is known that surfaces of silicate glasses hydrate when in water containing environments” are not so easy to understand.
  3. The abstract part should be more as encouragement of the potential readers, now it contains more information typical for the Conclusion part
  4. The introduction part should be definitely revised and should be more developed. There is a shortage of specific information.
  5. Lines 66-72 should be removed
  6. All chemical formula should be presented with subscripts.
  7. Lines 207-210 should be removed
  8. References should be revised (please look at 17, 18 and others); the same style should be presented.

Author Response

1)Remarks: In the whole text, ions should be presented as ions Sn4+ should be Sn4+ and many others.

Reply: Yes, the reviewer is right. We have modified all the valence states of ions into superscripts. All the revisions were highlighted in the manuscript.

Thanks the reviewer very much.

2)Remarks: Language presentation should be revised, English should be polished, and the part of sentences should be revised. In my opinion, the current language presentation in many cases seem to be too terse (lapidary). Part of sentences such as “It is known that surfaces of silicate glasses hydrate when in water containing environments” are not so easy to understand.

Reply: Thanks for the reviewer’s good suggestion. We have found a professional organization to modify the language and hope the language can meet the requirements.

3)Remarks: The abstract part should be more as encouragement of the potential readers, now it contains more information typical for the Conclusion part.

Reply: Thanks for the reviewer’s good comment. We have revised our abstract part. The information that should be in the conclusion part has been removed. The abstract after revision is as follows: “This work aims to explore the interaction between water and ion-exchanged aluminosilicate glass. The surface mechanical properties of ion-exchanged glasses after different hydration durations are investigated. The compressive stress and depth of stress layer are determined with a surface stress meter on the basis of photo-elasticity theory. The hardness and Young’s modulus are tested through nanoindentation. Infrared spectroscopy is used to determine the variation of surface structures of the glass samples. Results show that hydration has obvious effect on the hardness and Young’s modulus of the raw and ion-exchanged glasses. The hardness and Young’s modulus decrease to different extents after different hydration times, and the Young’s modulus shows some recovery with the prolonging of hydration time. The ion-exchanged glasses are more resistant to hydration. The tin side is more resistant to hydration than the air side. The results are expected to serve as reference for better understanding the hydration process of ion-exchanged glass.”

Thanks the reviewer very much.

4)Remarks: The introduction part should be definitely revised and should be more developed. There is a shortage of specific information.

Reply: Following the reviewer’s valuable suggestion, we have developed the introduction part and added specific information: “Several attempts have been made to study the influence of hydration on the mechanical properties of glasses [13–16]. The wear volume of fused quartz glass gradually increases with the increase in environment humidity; however, the wear tracks on soda–lime glass is smoother than those on fused quartz [13].  The surface wear of soda lime glass decreases with the increase in humidity, whereas that of sodium aluminosilicate glass remarkably increases [14]. Most of the studies focus on the variation of mechanical properties variation of the glasses, whereas studies on the mechanical properties of hydrated surfaces (especially at extremely low depth) are limited [3, 7, 14, 17]. Recent results obtained by nanoindentation show that the near-surface mechanical properties are altered by hydration, and the glass composition determining the resistance of glass is attacked with an aqueous solution. Hydration shows minimal effect on high durability glasses even at long immersion times, whereas hydration reduces the near-surface mechanical properties of low durability glasses remarkably [3]. Surface hydration also influences the residual compressive stress (CS) in ion-exchanged glasses [5, 18]. Alkali modifier ions on the surface of ion-exchanged glasses are leached when the ion-exchanged glasses were in contact with water regardless of surface compressive layer [3].”

Thanks the reviewer very much.

5)Remarks: Lines 66-72 should be removed

Reply: Thanks for the reviewer’s suggestion. Lines 66-72 have been removed from the manuscript.

6)Remarks: All chemical formula should be presented with subscripts.

Reply: We have modified all the chemical formulas into standard format.

Thanks the reviewer so much.

7)Remarks: Lines 207-210 should be removed

Reply: Thanks for the reviewer’s suggestion. Lines 207-210 have been removed.

8)Remarks: References should be revised (please look at 17, 18 and others); the same style should be presented.

Reply: Thanks for the reviewer’s suggestion. We have modified the references style. Thanks so much.

Reviewer 2 Report

Please, check English, the manuscript is difficult to read. Chemical formulas shoud be written using indexes as well as units „cm-1“. I suppose that a part of chemical description of the process is missing in the manuscript. The used method should be describe more detailed.

You still have in the manucritpt following “The Materials and Methods should be described with sufficient detailsto allow oth-66ers to replicate and build on the published results. Please note that the publication of your 67manuscript implicates that you must make all materials, data, computer code, and proto-68cols associated with the publication available to readers. Please disclose at the submission 69stage any restrictions on the availability of materials or information. New methods and 70protocols should be described in detail while well-established methods can be briefly de-71scribed and appropriately cited.” It is strange that among 6 authors noone notice it.

Who is the glass supplier?

The sentence  “The difference in composition and structure between the twosides can lead to differentproperties“ deserves reference https://doi.org/10.3390/ma14092218.

Author Response

Dear reviewer,

Thank you very much for your valuable comments and suggestions on our manuscript submitted to Materials (materials-1216179). We have carefully revised our manuscript according to the comments and suggestions. The following are the responses to the suggestions. We hope that our manuscript is now OK for Materials.

Yours sincerely

Dr. Xiaoyu Li

Response to the reviewers’ comments:

1)Remarks: Please, check English, the manuscript is difficult to read. Chemical formulas should be written using indexes as well as units „cm-1“. I suppose that a part of chemical description of the process is missing in the manuscript. The used method should be described more detailed.

Reply: Thanks for the reviewer’s good suggestion. We have found a professional organization to modify the language and we hope the language can meet the requirements now.

We have also modified all the chemical formulas and units into standard format, and all the revisions were highlighted in the manuscript.

We added some chemical description of the process in the introduction part: “The accepted hydrolysis and leaching reaction equations can be described as (the formulas cannot be shown in this blank, please see the attachment for details):

This process can be described by the classical stress corrosion theory developed by Charles and Hillig [11–12]. This theory depicts that molecules possessing proton donor sites and lone-pair orbitals such that the dissociation of the Si–O–Si bond is enhanced by water molecules through coupling across the Si–O bond to form an activated complex.”

The used method has been described more detailed in our article. We have added the size of our samples in the first paragraph of the materials and methods part: “The square plate glass samples measured an average of 50±0.5 mm on an edge and an average thickness of 4.0±0.02 mm”. The nanoindentation process was described more detailed in the paper: “Nanoindentation was conducted with a XP nanoindenter equipped with a Berkovich diamond indenter (Hysitron, TI 950). Prior to the nanoindentation tests, calibration was performed using a standard fused silica sample. The indentation load changed from 100 μN to 9000 μN in a proportional sequence, with a total of 36 indents. The Poisson’s ratio of the sample used in the calculation is 0.25. The method used in the experiments and calculations is based on ISO 14577 [28]. The hardness and modulus data were calculated using the well-known method developed by Oliver and Pharr (referred to as OP method).”

Thanks the reviewer so much.

2)Remarks: You still have in the manuscript following “The Materials and Methods should be described with sufficient details to allow others to replicate and build on the published results. Please note that the publication of your manuscript implicates that you must make all materials, data, computer code, and protocols associated with the publication available to readers. Please disclose at the submission stage any restrictions on the availability of materials or information. New methods and protocols should be described in detail while well-established methods can be briefly described and appropriately cited.” It is strange that among 6 authors no one notice it.

Reply: Thanks for the reviewer’s good suggestion. We are sorry for making such a mistake. Lines 66-72 and 207-210 have been removed from the manuscript.    

Thanks the reviewer very much.

3)Remarks: Who is the glass supplier?

Reply: The glass used was supplied by AVIC SANXIN. We have added this information in the manuscript.

Thanks for the reviewer’s good suggestion.

4)Remarks: The sentence “The difference in composition and structure between the two sides can lead to different properties” deserves reference https://doi.org/10.3390/ma14092218.

Reply: Thanks for the reviewer’s good suggestion. The reference https://doi.org/10.3390/ma14092218 is very valuable and we have added it into the manuscript as reference 24.

Thanks the reviewer so much.

Reviewer 3 Report

This is a study to understand the influence of hydration on the hardness and elastic modulus of ion-exchanged glasses. Results show the hydration can influence the glass surface mechanical properties. The driving force can be attributed to the formation of Si-OH structures. Results are interesting. The mechanical results appear to be supported by IR data. However, there are a few formatting related editing needed.

Example

Line 66 – 72; Line 207-210 are not needed.

Table 1 chemical formula need to follow standard format, such as subscripts ‘2’ needed for the SiO2.

IR wave number cm-1, the ‘-1’ superscript

Author Response

Dear reviewer,

Thank you very much for your valuable comments and suggestions on our manuscript submitted to Materials (materials-1216179). We have carefully revised our manuscript according to the comments and suggestions. The following are the responses to the suggestions. We hope that our manuscript is now OK for Materials.

Yours sincerely

Dr. Xiaoyu Li

Response to the reviewers’ comments:

1)Remarks: Line 66 – 72; Line 207-210 are not needed.

Reply: Thanks for the reviewer’s good suggestion. We are sorry for making such a mistake. Lines 66-72 and 207-210 have been removed from the manuscript.

Thanks so much.

2)Remarks: Table 1 chemical formula need to follow standard format, such as subscripts ‘2’ needed for the SiO2. IR wave number cm-1, the ‘-1’ superscript

Reply: Thanks for the reviewer’s good suggestion. We have modified all the chemical formulas and units into standard format.

Thanks so much.

Round 2

Reviewer 1 Report

The revised manuscript entitled “Insight into the interaction between water and ion-exchanged aluminosilicate glass by nanoindentation” submitted by Xiaoyu Li and co-workers for reconsideration for publication in the MDPI journal Materials presents definitely higher level than the first submission. The Authors have performed all the required corrections and add additional explanations. Also, additional explanations are properly presented in the answers to the Reviewer comments. Therefore, I consider the revised manuscript is suitable for publication in the MDPI journal Materials. My congratulations to the Authors. I wish You all the best and next good papers.

Reviewer 2 Report

The paper can be published.